# Complementary Feeding Practices and Parental Pressure to Eat among Spanish Infants and Toddlers: A Cross-Sectional Study

**DOI:** 10.3390/ijerph18041982

**Published:** 2021-02-18

**Authors:** Michelle Klerks, Sergio Roman, Maria Jose Bernal, Juan Francisco Haro-Vicente, Luis Manuel Sanchez-Siles

**Affiliations:** 1Hero Group, Research and Nutrition Department, 30820 Murcia, Spain; michelle.klerks@hero.es (M.K.); mjose.bernal@hero.es (M.J.B.); jfrancisco.haro@hero.es (J.F.H.-V.); 2Hero Group, Institute for Research and Nutrition, 5600 Lenzburg, Switzerland; 3Marketing Department, Facultad de Economía y Empresa, University of Murcia, 30100 Murcia, Spain; sroman@um.es

**Keywords:** complementary feeding practices, home-prepared food, parental pressure to eat, health, Spain

## Abstract

The introduction of complementary foods is a crucial stage in the development and determination of infants’ health status in both the short and longer-term. This study describes complementary feeding practices among infants and toddlers in Spain. Also, relationships among sample characteristics (both parents and their child), feeding practices (timing, type of complementary food), and parental pressure to eat were explored. Cognitive interviewing with 18 parents was used to refine the survey questions. Responses from a national random sample of 630 parents, who were responsible for feeding their infants and toddlers aged 3–18 months, were obtained. Solids, often cereals and/or fruits first, were introduced at a median age of five months. Fish and eggs were introduced around the age of nine and ten months. Almost all children were fed with home-prepared foods at least once per week (93%), and in 36% of the cases, salt was added. Interestingly, higher levels of parental pressure to eat were found in female infants, younger parents, parents with a full-time job, the southern regions of Spain, and in infants who were not fed with home-prepared foods. Our insights underline the importance of clear feeding recommendations that can support health care professionals in promoting effective strategies to improve parental feeding practices.

## 1. Introduction

Early childhood overweight and obesity represent a major health problem, particularly in developed countries [1,2], and Spain is no exception as evidenced in many studies over the years [3,4,5,6,7]. The current study provides new insights about complementary feeding practices of infants and toddlers in Spain. The introduction of complementary foods is a crucial stage in the development and determination of infants’ health status in both the short and longer-term [8,9,10,11]. Furthermore, the order of introduction, variety, and repeated exposure to complementary foods are related to the development of food preferences and eating habits later in life [12,13,14,15,16,17]. As a result, it is not surprising that: “the way in which a child is introduced to complementary foods may have effects on the individual’s entire life” [11] (p. 1).

Despite the importance of complementary feeding, there seem to be many perspectives regarding its “adequate” implementation. The World Health Organization (WHO) recommends exclusively breastfeeding infants up to six months of age and introducing complementary foods thereafter [18]. The European Food and Safety Authority (EFSA) and the European Society for Paediatric Gastroenterology, Hepatology and Nutrition (ESPGHAN) support the desirable goal to exclusively breastfeed until six months of age, as recommended by the WHO, however they elaborate on the possibilities to introduce complementary foods between the age of four and six months [19,20,21]. The EFSA recently concluded that no precise age for the start of complementary feeding can be determined, as this heavily depends on the infant’s characteristics and development [21]. In particular, they highlight that “Most infants do not need complementary foods for nutritional reasons up to around six months of age, with the exception of some infants at risk of iron depletion” (p. 5) and “that an infant might be developmentally ready for complementary foods before six months does not imply that this is necessary or desirable” (p. 5) [21]. As WHO considers infant formula as a complementary food, their recommendations are not directly comparable with those from EFSA and ESPGHAN, which exclude infant formula from complementary food [18,19,20,21]. Therefore, it is not surprising that a recent study [22] showed that in 31 of 38 European countries (82%) the introduction of complementary foods was recommended before six months. It was found that age recommendations differed in some countries depending on whether the infant is breastfed or formula-fed. In some countries (e.g., Germany and the UK) parents do not always wait for the introduction of foods until their infant reaches the recommended age [23,24,25]. In fact, consumption of solids before the recommended age can be found in any region globally, but the highest rates are found in Latin America and the Caribbean, and East Asia and the Pacific. In these regions, approximately 15% of infants are fed solids already when they are between two and three months old [26]. Accordingly, “the debate on the optimal age of solid food introduction is still open” [11] (p. 2) and “should be further investigated” [11] (p. 12). 

On a related issue, there is no consensus regarding the concrete order in which foods should be introduced [27], except for the recognized importance to complement the nutritional requirements of iron at six months of age [28]. Recommendations also seem to change over time. For instance, in Spain, it was recommended not to start with cereals with gluten, fish, and eggs until the age of eight to nine months, and legumes until 12 months [29], while currently, the recommendations are to introduce these foods, along with fruit, vegetables, meat, chicken and olive oil somewhere between 6–12 months [30]. More importantly, a gradual increase in variety and consistency of foods is regarded as essential from the beginning of complementary feeding [30,31]. Overall, differences in local culture seem to play a key role in shaping complementary feeding practices. 

Most of what we know until now about *what* and *when* infants and toddlers are fed up to two years of age, follows from observational studies administered in African developing countries [32,33,34,35,36], other non-Western countries like Vietnam and the United Arab Emirates [37,38,39], and developed countries such as the US and Canada [40,41]. Within Europe, infant feeding practices have been extensively investigated by observational studies in Scandinavian countries [42,43,44,45], the UK, Italy, Switzerland, Germany, the Netherlands, and France [23,25,46,47,48,49,50,51,52,53,54,55]. On the contrary, research about feeding practices in Spain has been mostly focused on children above two years of age and adolescents (e.g., [56,57,58]). Only a few studies have included infants and toddlers below 18 months, but they were focused on nutrient intake and adequacy [59,60,61,62,63]. 

Complementary feeding practices also involve specific behaviors parents apply to control *how much* their children eat. These behaviors relate to a parental feeding style (e.g., [64]). The pressure to eat, restriction of food, and the use of food as a reward are examples of them [65]. In this study, we are particularly interested in parental pressure to eat. Parental pressure to eat, in which parents do not respond to the child’s satiety signals and encourage their food intake [66], may cause children to be unable to regulate their own food intake [67]. Importantly, the pressure to eat “may have the unintended consequence of disrupting the development of intuitive and adaptive eating” [68] (p. 61) and has been associated with a tendency to overeat [69] and greater risk for overweight [70].

In summary, despite the general consensus about the pervasive importance of complementary feeding, there is an open debate in the academic community regarding *when*, *what*, and *how much* infants should eat in the complementary feeding stage. This study aims to provide a comprehensive description of feeding practices among Spanish infants and toddlers aged 3–18 months, including the timing of introduction, types of complementary food, and home-prepared feeding habits. In addition, we explore potential associations among parental and infant characteristics, feeding practices, and parental pressure to eat. A better understanding of these practices and their relationships can be used for further development of different approaches towards healthy complementary feeding in Spain and other similar developed countries.

## 2. Materials and Methods

### 2.1. Study Design and Participants

The current study analyzed data from a larger research project [71,72] about infants’ and toddlers’ eating behaviors, their nutritional status, and parents’ use of formula milk and complementary infant foods (e.g., cereals, baby jars, home-prepared foods). Data were collected in Autumn 2014. Different sizes of parent samples were used. Participants consisted of parents who: (1) had at least one child aged 3–18 months, (2) had primary responsibility for their infant feeding, and (3) their child did not have severe food allergies or chronic medical problems affecting their food intake. Ethical approval was obtained from the Research Ethical Committee of the University of Murcia.

Cross-sectional data were collected through an online consumer survey. A research firm collected the data and randomly selected a sample of Spanish parents whose infants and toddlers (aged 3–18 months) were representative for gender and Spanish Region from their online national panel. The initial sample consisted of 749 respondents. 34 cases were eliminated because of incomplete or inconsistent responses. 85 cases were not included in the data analyses of this study because parents did not feed their infants with complementary foods. The final sample consisted of 630 respondents.

### 2.2. Questionnaire

An initial version of the questionnaire was developed by the authors based on a literature review [41,51,73,74,75,76] along with feedback obtained from experts in nutrition and market research. The questionnaire was then tested among 18 parents using cognitive interviewing techniques. Most of them were female (88.9%), with a mean age of 34.3 years. Cognitive interviews require respondents to “think-aloud” or verbalize their thought process while completing the survey [77] (p. 287). This technique has been extensively used to test food and nutrition-related questionnaires (e.g., [75,78]).

The questionnaire was then distributed to a final pilot sample of 197 parents of children aged 3–18 months in Madrid, Barcelona, Sevilla, and Murcia. Trained interviewers randomly approached parents who were with their infant(s) in parks or at the entrance of the kindergarten. Most of the respondents were female (89.3%) and had a college degree (58.9%). Item means, frequencies, and alpha coefficients were calculated.

#### 2.2.1. Demographic Characteristics

Children demographic variables included age, gender, birth weight, position between brothers/sisters, and daycare attendance. Infant weight-for-age percentile was calculated using the WHO guidelines [79]. Parents demographics included age, gender, residence (both region and size of city), education, job intensity, monthly income, and marital status. 

#### 2.2.2. Complementary Feeding Practices

Parents were asked to indicate whether they were feeding their child with a selection of food categories (cereals, fruits, vegetables, yogurt, meat, cheese, fish, eggs, and legumes) and the age of the child at which each of these foods categories were introduced (timing). Intake frequency per food category per week was also asked on a 5-point interval (from “every day” to “rarely”). 

As for home-prepared feeding practices, parents answered who prepared it (mainly the parent, both the parent and someone else or mainly someone else), if salt was added while cooking it, type of food (puréed fruit, puréed vegetables, menu with meat and menu with fish) and intake frequency per week. 

#### 2.2.3. Parental Pressure to Eat

A three-item 5-point Likert scale (from 1 = “strongly disagree” to 5 = “strongly agree”), adapted from Birch et al. (2001) [74], was used to measure parental pressure to eat. One item of the original scale was found redundant as a result of the cognitive interviews and thus it was eliminated. 

### 2.3. Data Analysis

A descriptive analysis was carried out to obtain demographic characteristics, the timing of complementary feeding, intake frequencies of different food categories, and home-prepared feeding practices. Before analyses of quantitative data, normality was checked by the Kolmogorov-Smirnov test. Non-parametric data were presented in the median and interquartile range (Q3-Q1), while normal data as mean ± standard deviation (SD). Categorical variables are reported as percentages. The reliability of the multi-item scale was tested by calculating the composite reliability (CR). Research recommends cut-off values of 0.60 [80]. Two-tailed Pearson correlations and one-way analyses of variance (ANOVA) were carried out. Statistical analyses of data were performed using the Statistical Package for the Social Sciences (IBM SPSS Statistics for Windows, Version 25.0; Armonk, NY, USA; IBM Corp).

## 3. Results

### 3.1. Demographic Characteristics

The children’s and parents’ characteristics (*n* = 630) are described in Table 1. Slightly more than half of the children (54.6%) were toddlers between 12 and 18 months old, 51.3% of the total sample were boys and 38.9% attended the daycare. Parents, of which 79.5% were mothers, were on average 34.6 ± 4.2 years old. Most of the investigated families lived in a medium or big city (63.2%), primarily in the center of Spain (168 families, 26.7%). About three-quarters of the parents had a university degree and a quarter did not have a job. Almost all parents were married or lived together with a partner (94.4%).

### 3.2. Complementary Feeding Practices

#### 3.2.1. Timing of Introduction

The median age at which complementary foods were introduced was five months, with the first quartile at four months and the third quartile at six months. Of the total sample, 50 infants (8%) were introduced to solids at or during the age of three months. The majority of the infants that were introduced at or during the third month were offered cereals as their first complementary foods (39 infants, 78%). During the fourth and fifth month, 350 infants were introduced to solids (56%), 162 infants at six months (26%), and 68 (11%) beyond six months. 

Each food category was introduced at different age stages (Figure 1). Cereals (78%) and fruits (70%) were for most of the infants the first introduced solid foods at a median age of five months, followed by vegetables (32%) at a median age of six months, and yogurt (20%) and meat (15%) both at a median age of seven months. Fish, cheese, legumes, and eggs were the least chosen first solid foods. Fish was introduced at a median age of nine months, and cheese, eggs, and legumes at 9.5, 10, and 11 months, respectively (Figure 1).

#### 3.2.2. Frequency of Intake

The frequency of intake in which each food category was introduced per age group is depicted in Table 2. Almost half of the infants were fed with cereals every day in the age groups 7–11 months (46%) and 12–18 months (49%). In the age range of 3–6 months, cereals were given for 27% every day. Fruits and vegetables were given five to seven times a week around 90% of the cases in all age groups. Yogurt was more often given in the older age ranges. Between 7–11 and 12–18 months, yogurt was provided daily to 43% and 59% of the infants that received yogurt, respectively. Meat was mostly given three to four days a week between 7–11 months (36%) and between 12–18 months (45%). Fish was generally given three to four days a week: for around 45% in the age ranges 7–11 and 12–18 months. Eggs and legumes were not common to provide to infants (3–11 months), they were more likely to be given to toddlers (12–18 months). In particular, in this age group, both eggs (71%) and legumes (56%) were for the majority given one or two days a week (Table 2).

#### 3.2.3. Home-Prepared Feeding Practices

Of all children, 586 (93%) were given home-prepared foods at least once per week, whereas 7% were never fed home-prepared food (Table 3). The home-prepared foods were mainly prepared by the responsible parent (66%), occasionally it alternated between the responsible parent and someone else (21%), and for the minority, it was done by someone else (13%). For 36% (211 cases), salt was added to the home-cooked foods. In the age range from 3–6 months, salt was added in 13 cases (30%), from 7–11 months in 57 cases (26%), and in 141 cases from 12–18 months (43%).

As shown in Figure 2, in the age groups from 3–6 months and 7–11 months, more than half of the infants were fed with home-prepared puréed fruit every day, and for about a quarter five to six days a week. The percentages were slightly lower in toddlers from 12–18 months old, where simultaneously the percentage of toddlers that was rarely/never fed with fruit purées was higher (15%) (Figure 2).

Home-prepared puréed vegetables were less common to offer daily to children, as compared to home-prepared fruits: in 28%, 25%, and 17% of the children from 3–6, 7–11, and 12–18 months, respectively. The percentages of infants and toddlers that were rarely/never fed with home-prepared vegetables were higher as compared to fruits, ranging from 30–39%.

Approximately half of the infants between 3–6 months (49%) were rarely/never fed with meat. On the contrary, the percentage of children that were fed meat three to four days a week, increased from 14% in the first six months of life, to 33% and 43% in the subsequent age groups 7–11 and 12–18 months, respectively. Most of the infants between 3–6 months were rarely/never fed with fish (70%). However, this percentage was halved in infants of 7–11 months of age, and close to zero (6%) in toddlers of 12–18 months of age. The percentages of toddlers (12–18 months) fed with fish were remarkably high, with 64% of them being fed with fish three or more times a week (Figure 2).

### 3.3. Pressure to Eat

The reliability value for the pressure to eat scale was acceptable (CR = 0.75). Overall, the pressure to eat was rated by the parents as moderately high (3.46 ± 0.82). Values for the three items are shown in Table 4.

### 3.4. Associations between Sample Characteristics, Feeding Practices and Pressure to Eat

It was tested whether sample characteristics (both characteristics of the children and the responsible parents, as explained in Table 1) influenced the timing of the introduction of complementary foods. None of the characteristics affected the age of introduction to solids (all *p* > 0.05), except for the parents’ gender (F_1,628_ = 11.83; *p* < 0.01). Fathers started on average 0.6 months later with the introduction of solids (5.65 ± 2.28 months) as compared to mothers (5.07 ± 1.53 months). 

Furthermore, possible relevant associations between sample characteristics and parental pressure to eat were investigated. Higher levels of pressure to eat were related to lower infant’s birth weight (r = −0.10, *p* = 0.01) and lower current weight percentiles (r = −0.10, *p* = 0.01). Higher pressure to eat ratings were found in female infants (F_1,628_ = 5.67, *p* = 0.02), parents below the age of 30 years (F_1,628_ = 3.68, *p* = 0.05), and full-time workers (F_1,483_ = 8.55, *p* < 0.01). Also, there were significant differences in pressure to eat among the several regions in Spain (F_6,623_ = 2.81, *p* = 0.01). In the South of Spain, the pressure to eat was rated the highest (3.57 ± 0.82) and post-hoc LSD showed that pressure to eat in the South significantly differed from the Canary Islands (mean difference 0.61 ± 0.28), the North (mean difference 0.37 ± 0.13) and the North-West (mean difference 0.31 ± 0.13) of Spain. Significant associations tested with one-way ANOVA between sample characteristics and pressure to eat are shown in Table 5. The pressure to eat was not significantly associated with the infants’ age, position between brothers and sisters, daycare attendance, parents’ gender, city size, educational level, monthly income, and marital status (all *p* > 0.05).

Even though differences were not significant, the pressure to eat was higher in parents with primary education (mean value = 3.55), as compared to those with secondary education (mean value = 3.46) and a university degree (mean value = 3.45). Similarly, the pressure was also faintly higher in the group with income lower than 1000€ (mean value = 3.48) as compared to the group with incomes higher than 1000€ (mean value = 3.47). 

Pressure to eat was not associated with the timing of introduction to solids. However, the pressure to eat was associated with the introduction of some complementary foods (fruits, vegetables, and meat) in infants and toddlers of 6–18 months old. Higher values of pressure to eat were found in those children that were not introduced to these three food types (Table 6).

Interestingly, the pressure to eat tended to take place less (F1,628 = 3.83; p = 0.05) when children were exposed to any home-prepared foods (3.45 ± 0.82) compared to children that were not exposed to any home-prepared foods (3.70 ± 0.83). 

## 4. Discussion

This study described complementary feeding practices and explored potential associations among parental and infant characteristics, feeding practices, and parental pressure to eat among Spanish infants and toddlers. Several important insights can be derived from our results.

First, we observed that the majority of the parents started with the introduction of complementary foods between four and six months old, which is consistent with European [19,20] and Spanish recommendations [29,30] and findings from Costantini et al. (2018) [24] in the UK and Italy. Similar to the results obtained in Norway [42], Denmark [44], Switzerland [47], and the Netherlands [55], solids were rarely given before the age of four months (8%). On a related issue, early introduction (before the age of four months) in prior research has been associated with a younger maternal age [23,47,49,53,54,55,81] and a lower educational level [53,54,55,81,82]. Our results could not confirm an association between timing of introduction and parental age nor educational level. We found that mothers tend to introduce solids at an earlier age than fathers. To our knowledge, this finding has not been investigated or found in previous research yet. There has been little research on the role of the father in infant feeding, as often only mothers are included in the analyses [83]. This is unfortunate since fathers also play a key role in the development and feeding habits of their children [84,85].

As for the types of first introduced solids, our findings were consistent with an earlier study conducted in Spain [76]. In particular, cereals along with fruits were the most common first given solids. This is also in line with evidence from other countries such as UK, Ireland, and Canada [40,48,50,53]. In contrast, Turkish infants were fed yogurt first [86], and studies in France [51,52,87] and Germany [87] showed that vegetables were also a prominent food category to start with. It is striking that yogurt was introduced around the age of seven months, while it is advised in Spain to postpone the introduction until the age of nine months [29,30]. In addition, this study showed that Spanish parents wait for the introduction of fish and eggs to a later age stage. Indeed, these were the Spanish recommendations in the past [29], but current European recommendations are to introduce potentially allergenic foods (e.g., fish, eggs, peanut) no later than other solids [21]. Parents might be a bit reluctant to provide those foods due to the changeability of feeding guidelines that leads to confusion, the use of multiple sources of information, or simply concerned about the infant’s readiness for certain foods [83].

Our findings were consistent with the Spanish recommendations for the frequency of food intake of cereals, fruit, vegetables, meat, and fish [29,30], although not always exact amounts are recommended. Given the fact that breast milk or infant formula are still the most important dairy sources during the complementary feeding period, the consumption frequency of yogurt might need some attention [29,30]. In particular, the ENALIA study (2017) found a similar intake of yogurt, as 41.6% and 56.1% of Spanish children between 6–11 months and 12–35 months consumed yogurt once a day, respectively [61]. According to the IFS survey, 68.0% of British infants between 8–10 months of age consumed yogurt at least once a day [48]. A high yogurt intake along with an adequate milk consumption could lead to a higher than desirable protein intake. An excessive intake of protein in infancy was found in previous studies [59,88,89] and has been related to higher levels of overweight and obesity in later life [90]. 

Almost all children were fed with home-prepared foods at least once a week, where puréed fruit was in each age category most frequently prepared. One plausible explanation for this could be that fruit purées are easier to prepare as compared to meat and fish purées. An important issue in preparing infant food at home relates to the addition of salt. Salt addition is discouraged in infants up to two years of age; not only because they are too immature and unable to excrete an overload of salt [20,88], but also to prevent encouragement of unhealthy preferences [20,91]. Contrary to the recommendations to avoid the addition of salt, especially during the first year of life [30], a third of the parents surveyed in our study admitted adding salt when cooking at home. This percentage was lower (11%) in Spanish infants according to the ENALIA study (2017) [61]. 

Consistent with previous evidence [92,93,94,95,96], the pressure to eat was negatively correlated to the child’s current weight percentile. Because of the cross-sectional design of our study, the direction of the relationship cannot be established. It could well be that it concerns a vicious circle. Parents might react to the weight of the child by pressuring them to eat. This supports the thought that parental feeding practices are a result as well as a predictor of child weight, where a pressuring feeding style is counterproductive [97,98]. In addition, we found that pressure to eat was negatively associated with the infants’ birth weight. The same associations for birth weight were found earlier with a pressuring feeding style in a cross-sectional study [99]. These observations may imply that parental pressure to eat is somehow already developed soon after the child is born and thus may suggest that it is a result rather than a predictor of child weight. A noteworthy outcome of the present study that needs to be stressed out is that pressuring feeding practices were the highest in the South of Spain and differed significantly from the North and North-West of Spain. Interestingly, higher rates of childhood obesity can be found in southern areas of Spain, as compared to the northern areas [100]. This may imply that, in line with the study by McPhie et al. (2012) [101], the pressure to eat in early life could lead to weight gain later on. 

Our study adds to the literature by identifying other relevant variables related to the pressure to eat. Unlike prior research that did not find differences in infant’s or children’s gender [99,102,103,104], we found that pressure to eat was higher in female infants. As parental pressure to eat has been shown to be positively correlated to their concern about their children being underweight [94], we could argue that parents might be more concerned about girls becoming underweight and thereby putting more pressure on them than on boys. Still, further studies are needed in this regard. Furthermore, parental pressure was higher among younger parents, which is consistent with findings from Brown and Lee (2013) [105] in the UK that showed that maternal age was inversely associated with an encouraging feeding style. We also found that parents working full-time exerted a major pressure to eat, as compared to those working part-time. Lack of time and impatience might be a plausible explanation for this finding, but little is known about the potential role parental employment plays in infant complementary feeding practices. Importantly, we found that infants’ exposure to home-prepared meals tended to be associated with lower pressure to eat. In Synnott et al. (2007) [106], mothers explained that they prepared meals at home to ensure their baby would like the food. Even though the pressure to eat was only slightly higher in lower-income groups and with lower levels of education, our differences were not significant in line with prior research [101,107,108,109,110]. Still, other studies have found significant differences [111,112,113,114,115,116]. Given these inconsistent findings, we encourage researchers to deepen into the relationship of pressure to eat, parents’ income, and educational level.

There are strengths and limitations to this study that need to be considered. The present study is strengthened by the fact that a nationally representative sample was gathered, as 630 final responses distributed over each region in Spain were included in the analyses. The sample consisted of both mothers and fathers which provides new insights about fathers’ feeding practices and how these differ from mothers’ feeding practices. Furthermore, before conducting the online survey, the questionnaire was tested and validated where after the content was checked and modified. However, it has been shown previously that observed and reported parental feeding styles and practices might not always be in congruence [66,117,118]. Therefore, it must be noted that there was a chance of reporting bias. Also, because the current study used a cross-sectional approach, it is difficult to make causal inferences since variables were measured at one single point in time. Thus, longitudinal studies are recommended to establish the direction of relations between infant weight (trajectories) and parental pressure to eat. Finally, future research on infant feeding practices could include fathers too and examine other determinants of parental pressure to eat.

## 5. Conclusions

Complementary feeding practices were identified and explored in parents with infants and toddlers from 3–18 months in Spain. In line with the European and Spanish recommendations, the timing of introduction to solids occurred generally between four and six months of age. Fathers seem to introduce solids later than mothers. Comparable to many other countries, cereals and fruits were the most common first given solids. Parents provide yogurt earlier than recommended, while they wait for the introduction of fish and eggs. Frequent consumption of yogurt in addition to adequate milk consumption could be worrisome due to possibly excessive levels of protein intake. The high prevalence of salt added while preparing home-made complementary food needs to be addressed. Interestingly, higher levels of parental pressure to eat were found in female infants, younger parents, parents with a full-time job, the southern regions of Spain, and in infants that were not fed with home-prepared foods.

Hopefully, our insights underline the importance of clear and more specific feeding recommendations that can support health care professionals and parents in promoting effective strategies to improve parental feeding practices. In particular, national and regional guidelines would have to continuously incorporate the latest scientific evidence from experts’ bodies (e.g., EFSA, ESPGHAN) in key areas such as the right moment to introduce allergens (e.g., fish and eggs) in infants’ diet. Health care professionals may advise parents, particularly younger and less experienced ones working full-time, about the potential problems associated with pressuring infants to eat (e.g., eating disorders, risk of overweight). At a community level, educational initiatives that stress public awareness about when, what and how to feed babies are encouraged through communication campaigns and tools that instruct parents, for example, on how to make adequate nutritional and healthy adaptations of their traditional/cultural recipes for their infants.

## Figures and Tables

**Figure 1 ijerph-18-01982-f001:**
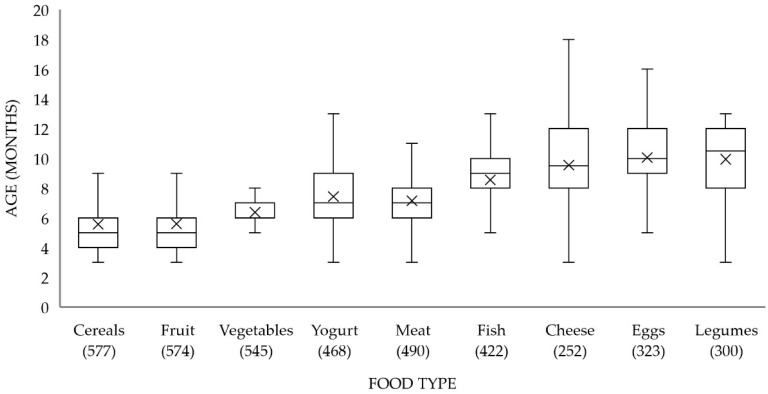
The age of introduction of each food category into the Spanish infants’ diets. The line in the boxplots represents the median and the cross represents the mean. The number between parentheses is the number of infants from the total sample that had already been introduced to the corresponding food category.

**Figure 2 ijerph-18-01982-f002:**
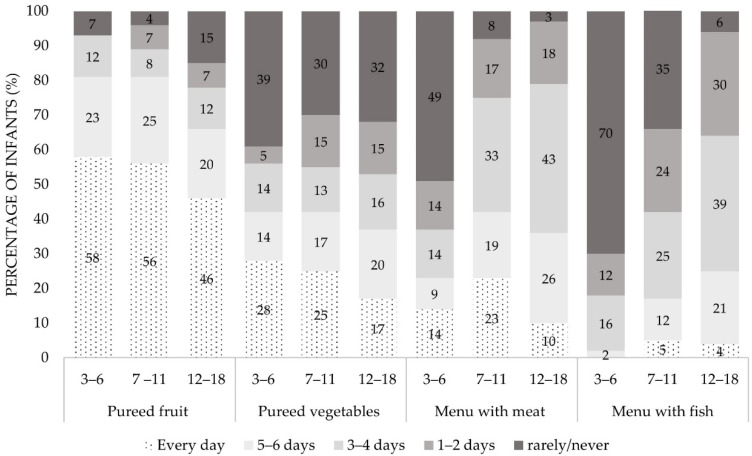
Frequency of intake of home-prepared purées and menus per week. Total *n* = 586, 3–6 months: *n* = 43, 7–11 months: *n* = 216, 12–18 months: *n* = 327. To specify rarely/never means a consumption with a lower frequency than 1 time per week or even no consumption at all. Percentages are rounded.

**Table 1 ijerph-18-01982-t001:** Demographic characteristics, *n* = 630.

	*n* or Mean ± SD	Percentage (%)
Characteristics of infants and toddlers		
Age (months)	11.8 ± 3.6	
3–6 months	56	8.9
7–11 months	230	36.5
12–18 months	344	54.6
Gender		
Boy	323	51.3
Girl	307	48.7
Birth weight (g)	3222 ± 493	
Position between brothers/sisters		
First	39	6.2
Second	207	32.9
Third	32	5.1
Fourth	5	0.8
No brothers and/or sisters	347	55.1
Daycare attendance		
Yes	245	38.9
No	385	61.1
More than three days a week lunch at daycare		
Yes	172	27.3
No	73	11.6
Characteristics of parents		
Age (years)	34.6 ± 4.2	
Gender		
Female	501	79.5
Male	129	20.5
Region		
North-East	121	19.2
East	90	14.3
South	124	19.7
Center	168	26.7
North-West	63	10.0
North	55	8.7
Canary Islands	9	1.4
City size		
Medium/big city	398	63.2
Small city/village/countryside	232	36.8
Education		
Primary school	15	2.4
Secondary school	146	23.2
University bachelor/master/PhD	469	74.4
Job intensity		
Full-time	378	60.0
Part-time/per hours	107	17.0
Unknown/not working	145	23.0
Total monthly income (€)		
<1000	43	6.9
>1000	491	77.9
Do not know/no answer	96	15.2
Marital status		
Divorced/single	35	5.6
Married/living with a partner	595	94.4

**Table 2 ijerph-18-01982-t002:** Frequency of intake of solid foods per week per age group. Percentages are calculated from the sample that was introduced to the corresponding food category. To specify rarely means a consumption with a lower frequency than 1 time per week.

Food Categories Per Age	*n*	Every Day	5–6 Days	3–4 Days	1–2 Days	Rarely
Cereals	577					
3–6 months	51	14 (27%)	9 (18%)	6 (12%)	2 (4%)	20 (39%)
7–11 months	206	94 (46%)	21 (10%)	16 (8%)	12 (6%)	63 (30%)
12–18 months	320	157 (49%)	33 (10%)	22 (7%)	16 (5%)	92 (29%)
Fruits	574					
3–6 months	36	25 (69%)	9 (25%)	2 (6%)	0 (0%)	0 (0%)
7–11 months	211	167 (79%)	30 (14%)	10 (5%)	4 (2%)	0 (0%)
12–18 months	327	246 (75%)	65 (20%)	13 (4%)	3 (1%)	0 (0%)
Vegetables	545					
3–6 months	25	16 (64%)	7 (28%)	2 (8%)	0 (0%)	0 (0%)
7–11 months	199	150 (75%)	31 (16%)	13 (7%)	5 (2%)	0 (0%)
12–18 months	321	216 (67%)	68 (21%)	25 (8%)	10 (3%)	2 (1%)
Yogurt	468					
3–6 months	15	4 (27%)	6 (40%)	2 (13%)	3 (20%)	0 (0%)
7–11 months	159	68 (43%)	46 (29%)	27 (17%)	12 (7%)	6 (4%)
12–18 months	294	174 (59%)	62 (21%)	40 (14%)	13 (4%)	5 (2%)
Meat	490					
3–6 months	13	4 (31%)	4 (31%)	4 (31%)	1 (8%)	0 (0%)
7–11 months	179	50 (28%)	43 (24%)	64 (36%)	19 (10%)	3 (2%)
12–18 months	298	46 (15%)	82 (28%)	133 (45%)	36 (12%)	1 (0%)
Cheese	252					
3–6 months	3	0 (0%)	1 (33%)	1 (33%)	0 (0%)	1 (33%)
7–11 months	35	1 (3%)	4 (11%)	10 (29%)	13 (37%)	7 (20%)
12–18 months	214	14 (6%)	41 (19%)	53 (25%)	70 (33%)	36 (17%)
Fish	422					
3–6 months	5	0 (0%)	2 (40%)	3 (60%)	0 (0%)	0 (0%)
7–11 months	121	6 (5%)	19 (16%)	53 (44%)	37 (31%)	6 (5%)
12–18 months	296	15 (5%)	58 (20%)	134 (45%)	85 (29%)	4 (1%)
Eggs	323					
3–6 months	4	0 (0%)	1 (25%)	1 (25%)	1 (25%)	1 (25%)
7–11 months	61	0 (0%)	2 (3%)	6 (10%)	39 (64%)	14 (23%)
12–18 months	258	1 (0%)	6 (2%)	33 (13%)	183 (71%)	35 (14%)
Legumes	300					
3–6 months	4	0 (0%)	0 (0%)	1 (25%)	1 (25%)	2 (50%)
7–11 months	51	1 (2%)	6 (12%)	10 (20%)	27 (53%)	7 (14%)
12–18 months	245	1 (1%)	15 (6%)	56 (23%)	138 (56%)	35 (14%)

**Table 3 ijerph-18-01982-t003:** Home-prepared feeding practices per age group.

Age Group	Home-Prepared Food, *n* (%)	Never Home-Prepared Food, *n* (%)
3–6 months	43 (77%)	13 (23%)
7–11 months	216 (94%)	14 (6%)
12–18 months	327 (95%)	17 (5%)
Total	586 (93%)	44 (7%)

**Table 4 ijerph-18-01982-t004:** Descriptive statistics of pressure to eat scale and its items (*n* = 630).

Pressure to Eat Scale and Its Items ^1^	Mean ± SD
My child should always eat all of the food	3.45 ± 1.13
I try my best to make sure my child eats enough	4.01 ± 0.88
My child would eat much less if I do not pay attention to the feeding	2.93 ± 1.20

^1^ Items scored 1–5 using a 5-point Likert scale, with 1 = “strongly disagree”, 2 = “disagree”, 3 = “neither agree nor disagree”, 4 = “agree” and 5 = “strongly agree”.

**Table 5 ijerph-18-01982-t005:** One-way ANOVA testing for the effects of sample characteristics on the pressure to eat. Only significant results are shown.

Variable	Characteristics		Mean ± SD	F-Value	*p*-Value
Pressure to eat	Gender infant	Girl	3.54 ± 0.84	F_1,628_ = 5.67	0.02
		Boy	3.39 ± 0.81		
	Parent’s age	≤30 years	3.61 ± 0.78	F_1,628_ = 3.68	0.05
		>30 years	3.44 ± 0.83		
	Job intensity	Full-time	3.50 ± 0.79	F_1,483_ = 8.55	<0.01
		Part-time/per hours	3.25 ± 0.76		
	Region	North	3.20 ± 0.81	F_6,623_ = 2.81	0.01
		North-East	3.47 ± 0.78		
		East	3.56 ± 0.81		
		South	3.57 ± 0.82		
		Center	3.51 ± 0.77		
		North-West	3.26 ± 0.95		
		Canary Islands	2.96 ± 1.20		

**Table 6 ijerph-18-01982-t006:** One-way ANOVA testing for the effects of the introduction of several food categories on the pressure to eat in infants and toddlers from 6–18 months old, *n* = 580. Only significant results are shown.

Variable	Food Category	Child Does Not Eat This Food Mean ± SD	Child Eats This Food Mean ± SD	F-Value	*p*-Value
Pressure to eat	Fruit	3.88 ± 0.69	3.44 ± 0.83	F_1,578_ = 4.22	0.04
	Vegetables	3.74 ± 0.72	3.43 ± 0.84	F_1,578_ = 4.78	0.03
	Meat	3.64 ± 0.80	3.42 ± 0.83	F_1,578_ = 5.37	0.02

## Data Availability

The data presented in this study are available on request from the corresponding author.

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
