# Peer review of "Complementary Feeding Practices and Parental Pressure to Eat among Spanish Infants and Toddlers: A Cross-Sectional Study"

_ijerph, 2021, doi:10.3390/ijerph18041982_

Round 1
Reviewer 1 Report
This cross-sectional study describes complementary feeding practices and parental pressure to eat among Spanish infants and toddlers. The authors provide comprehensive literature review and summarize and discuss the results of the study in a clear and easy to understand language.
The authors also acknowledge the strength and limitations of the study while highlighting important observations (e.g., fathers’ role in the feeding children, parental pressure to eat being most common among female parents and on female infants; weight of babies being the signal, not the result of parental pressure to eat at the early stages of child growth, and the fact the practice being more common in the southern Spain where higher rates of childhood overweight/obesity exist, etc.) Overall, the study provides valuable insights to program/policy makers around feeding recommendations and guidelines in the region.
One area this reviewer felt lacking in the literature review section is on the issue of why there’s disparity between the WHO complementary feeding guidelines and the Spanish and European versions, specially on the timing of the introduction of complementary feeding. Some reflection on this in your introduction section would be a plus.
Author Response
This cross-sectional study describes complementary feeding practices and parental pressure to eat among Spanish infants and toddlers. The authors provide comprehensive literature review and summarize and discuss the results of the study in a clear and easy to understand language.
The authors also acknowledge the strength and limitations of the study while highlighting important observations (e.g., fathers’ role in the feeding children, parental pressure to eat being most common among female parents and on female infants; weight of babies being the signal, not the result of parental pressure to eat at the early stages of child growth, and the fact the practice being more common in the southern Spain where higher rates of childhood overweight/obesity exist, etc.) Overall, the study provides valuable insights to program/policy makers around feeding recommendations and guidelines in the region.
Our response: We would like to thank you for taking the time to review our paper and providing your constructive and positive feedback.
One area this reviewer felt lacking in the literature review section is on the issue of why there’s disparity between the WHO complementary feeding guidelines and the Spanish and European versions, specially on the timing of the introduction of complementary feeding. Some reflection on this in your introduction section would be a plus.
Our response: This is a fair point. We have tried to provide clearer information by incorporating the following issues in the revised version:
- EFSA and ESPGHAN support the desirable goal to exclusively breastfeed until 6 months of age as recommended by the WHO. However, both EFSA and ESPGHAN elaborate on the possibilities to introduce complementary foods between the age of 4 and 6 months, depending on the infant’s specific needs.
- The definitions by the WHO and EFSA/ESPGHAN of what entails the term “complementary feeding” are formulated differently. The WHO describes the complementary feeding period as “the period during which other foods or liquids are provided along with breast milk” and states that “any nutrient-containing foods or liquids other than breast milk given to young children during the period of complementary feeding are defines as complementary foods”, whereas the ESPGHAN and EFSA use the term “complementary feeding” to embrace all solid and liquid foods other than breast milk or infant formula/follow-on formula or both.
- Given the above facts, it is not surprising that implementation of the age of introduction of complementary foods differs between countries.
Please see new text in red letters in pp.1-2 (up to line 59):
Despite the importance of complementary feeding, there seems to be many perspectives regarding its “adequate” implementation. The World Health Organization (WHO) recommends to exclusively breastfeed infants up to six months of age and to introduce complementary foods thereafter [18]. The European Food and Safety Authority (EFSA) and the European Society for Paediatric Gastroenterology, Hepatology and Nutrition (ESPGHAN) support the desirable goal to exclusively breastfeed until 6 months of age, as recommended by the WHO, however they elaborate on the possibilities to introduce complementary foods between the age of 4 and 6 months [19-21]. The EFSA recently concluded that no precise age for the start of complementary feeding can be determined, as this heavily depends on the infant’s characteristics and development [21]. In particular, they highlight that “Most infants do not need complementary foods for nutritional reasons up to around six months of age, with the exception of some infants at risk of iron depletion” (p. 5) and “that an infant might be developmentally ready for complementary foods before six months does not imply that this is necessary or desirable” (p. 5). As WHO considers infant formula as complementary food, their recommendations are not directly comparable with those from EFSA and ESPGHAN, which exclude infant formula from complementary food [18–21]. Therefore, it is not surprising that a recent study [22] showed that in 31 of 38 European countries (82%) the introduction of complementary foods was recommended before six months. Also, it was found that age recommendations differed in some countries depending on whether the infant is breastfed or formula-fed.
Reviewer 2 Report
Complementary feeding practices and parental pressure to eat among Spanish infants and toddlers: A cross-sectional study.
In the present study, the authors provide detailed information about the most common practices to introduce complementary feeding in infants and toddlers in Spain.
Main concerns:
-As described, the methodology is very hard to follow. I would recommend streamlining and clarifying the different stages of the project.
-I find the conclusions a bit lacking. The authors expressed that they hope their findings would inform formal feeding recommendations, but do not elaborate. Could the authors propose some such recommendations?
Author Response
Complementary feeding practices and parental pressure to eat among Spanish infants and toddlers: A cross-sectional study. In the present study, the authors provide detailed information about the most common practices to introduce complementary feeding in infants and toddlers in Spain.
Our response: We would like to thank you for taking the time to review our paper and providing your valuable feedback which we address below.
Main concerns:
-As described, the methodology is very hard to follow. I would recommend streamlining and clarifying the different stages of the project.
Our response: Several changes have been made in the methodology section to make it clearer and easier to follow (please see changes in red letters in the revised manuscript on p.3 lines 106-156). In particular, the “materials and methods” section is now splitted into 3 subsections, and not 4 like before. Subsection 2.1. is now entitled “Study design and participants” and include information that was previously located in subsection 2.4 (which has now been eliminated). Subsection 2.2. is now entitled “Questionnaire” and new subheadings have been incorporated in this subsection to make it clear how variables have been measured. Also, the paragraph starting “During the interviews….Interviews lasted on average 50 minutes” has been eliminated.
On a final note, somehow while preparing our manuscript for its first submission we made a mistake and did not include all information regarding the pretest of the questionnaire, which was conducted in 4 Spanish cities (including Murcia) with 197 parents, and not only in Murcia. We do apologize for this mistake. The revised version already includes the correct information (lines 131-135).
-I find the conclusions a bit lacking. The authors expressed that they hope their findings would inform formal feeding recommendations, but do not elaborate. Could the authors propose some such recommendations?
Our response: Yes, this is indeed a fair point. Accordingly, the text shown below (changes in red letters) has been incorporated to the manuscript in our conclusions (please see p. 12, lines 413-424):
Hopefully, our insights underline the importance of clear and more specific feeding recommendations that can support health care professionals and parents in promoting effective strategies to improve parental feeding practices. In particular, national and regional guidelines would have to continuously incorporate the latest scientific evidence from experts’ bodies (e.g., EFSA, ESPGHAN) in key areas such as the right moment to introduce allergens (e.g., fish and eggs) in infants’ diet. Health care professionals may advice parents, particularly younger and less experienced ones working full-time, about the potential problems associated with pressuring infants to eat (e.g., eating disorders, risk of overweight). At a community level, educational initiatives that stress public awareness about when, what and how to feed babies are encouraged through communication campaigns and tools that instruct parents, for example, on how to make adequate nutritional and healthy adaptations of their traditional/cultural recipes for their infants.
Reviewer 3 Report
Thank you for the opportunity to review this interesting manuscript. The authors are to be commended for preparing a well-written and comprehensive document. A number of associations between parental feeding practices and pressure to eat were examined and authors discussed findings in a thoughtful manner. A few issues to consider:
- Please say more about the practical / clinical relevance of the current findings. While the authors state that findings will be helpful for health care professionals, a few sentences on how these findings may actually be applied may will be an important addition.
- It is surprising to me that more of the demographic characteristics (income, in particular) were not associated with pressure to eat. Is it possible that sampling procedures were not inclusive of the entire range of potential participants?
Author Response
Thank you for the opportunity to review this interesting manuscript. The authors are to be commended for preparing a well-written and comprehensive document. A number of associations between parental feeding practices and pressure to eat were examined and authors discussed findings in a thoughtful manner. A few issues to consider:
- Please say more about the practical / clinical relevance of the current findings. While the authors state that findings will be helpful for health care professionals, a few sentences on how these findings may actually be applied may will be an important addition.
Our response: Yes, this is indeed a good point already highlighted by other reviewer. Accordingly, the text shown below (changes in red letters) has been incorporated to the manuscript in our conclusions (please see p. 12, lines 413-424):
Hopefully, our insights underline the importance of clear and more specific feeding recommendations that can support health care professionals and parents in promoting effective strategies to improve parental feeding practices. In particular, national and regional guidelines would have to continuously incorporate the latest scientific evidence from experts’ bodies (e.g., EFSA, ESPGHAN) in key areas such as the right moment to introduce allergens (e.g., fish and eggs) in infants’ diet. Health care professionals may advice parents, particularly younger and less experienced ones working full-time, about the potential problems associated with pressuring infants to eat (e.g., eating disorders, risk of overweight). At a community level, educational initiatives that stress public awareness about when, what and how to feed babies are encouraged through communication campaigns and tools that instruct parents, for example, on how to make adequate nutritional and healthy adaptations of their traditional/cultural recipes for their infants.
- It is surprising to me that more of the demographic characteristics (income, in particular) were not associated with pressure to eat. Is it possible that sampling procedures were not inclusive of the entire range of potential participants?
Our response: We are glad you brought this to our attention. We are using a random sample of Spanish parents whose children are representative for gender and Spanish Region. We were somehow expecting family income and also parents’ education to be negatively related to pressure to eat. Still, prior evidence in this regard is not consistent. We only found a couple of studies reporting the relationship between pressure and family income. Dinkevich et al. (2015) found it to be significant among a sample of 169 mothers (88% African American), whereas McPhie et al. (2012) found no significant link in a sample of 117 Australian mothers. In both cases, sample size was pretty limited. Upon reflection, and based on our own experience conducting surveys, we have learnt that people are very reluctant to reveal their (true) income. This is why we left the possibility of parents not disclosing this information in our study. We have conducted additional analyses splitting the sample into new groups. Our results show that pressure was only marginally higher (mean value=3.48) when income was below 1000€ as compared to income above 1000€ (mean value=3.47).
On a related issue, we were also intrigued as to why education and pressure were not related. Similarly to income, existing evidence here is not conclusive, yet this relationship has received more attention. Many studies have found it to be significant (Dinkevich et al., 2015; Gonçalves et al., 2017; Gross et al., 2010; Loth et al., 2013, Real et al., 2014, Russell et al., 2018), whereas others found no relationship (Duke et al., 2004; Li et al., 2014; McPhie et al., 2012; Timby et al., 2014; Warkentin et al., 2018). We have conducted new analyses disaggregatting education into three levels (in the first version we only considered two levels). Interestingly, we found that pressure was higher in the lower the education (primary school; mean value=3.55), as compared to the other ones, but differences were not significant.
A summary of these considerations and analyses is now provided in footnote number 1 (p. 9) and lines 379-384 on p. 11. Also, new references are provided in the revised version of the manuscript (refs 108-117).